# Role of Magnesium in Ultra-Low-Radioactive Titanium Production for Future Direct Dark Matter Search Detectors

**DOI:** 10.3390/ma15248872

**Published:** 2022-12-12

**Authors:** Marina Zykova, Elena Voronina, Alexander Chepurnov, Dmitry Rymkevich, Aleksey Tankeev, Sergey Vlasov, Alexander Chub, Igor Avetissov

**Affiliations:** 1Department of Chemistry and Technology of Crystals, Mendeleev University of Chemical Technology, 125047 Moscow, Russia; 2Skobeltsyn Institute of Nuclear Physics, Lomonosov Moscow State University, 119234 Moscow, Russia; 3Belgorod State National Research University, 308015 Belgorod, Russia; 4“AVISMA” Branch of Public Stock Company “VSMPO-AVISMA Corporation”, 618421 Berezniki, Russia; 5ARMOLED Ltd., 125047 Moscow, Russia

**Keywords:** magnesium, pure substance, inductively coupled plasma mass spectrometry, Kroll process, vacuum sublimation

## Abstract

Ultra-low-radioactive titanium is the main perspective material for cryostat fabrication in dark matter search experiments. The pathways of the uranium and thorium contamination of Ti sponges produced by the Kroll process were analyzed. The general role of Mg in Ti sponge contamination by U and Th was established. It was found that when transformed to MgCl_2_ in the Kroll process, Mg was purified from U and Th, and further MgCl_2_ reduction and sublimation makes it possible to produce low-radioactive Ti sponges.

## 1. Introduction

Recent astrophysical and cosmological measurements reveal that ordinary barionic matter makes less than 5% of the total mass of the observable universe manifested in gravitational interactions, while the nature of the remaining 95% of mass remains unknown. The direct detection of dark matter (DM) in collisions with barionic matter is of fundamental importance for cosmology, astrophysics, and elementary particle physics [1]. One of the hypotheses predicts that DM could be formed by stable particles that have non-zero mass and can presumably experience weak interactions. The weakly interactive massive particles (WIMPs) could be detected through their elastic scattering on target nuclei and electrons. Low-energy (<100 keV) recoils are produced as a result of this process and they are registered by the experimental setup. It means that all possible radioactive backgrounds, which could mimic WIMP- recoil processes, should be suppressed as much as possible. A special area of experimental science—low-radioactive technology—appeared to serve such experiments, and a lot of efforts worldwide are being spent to develop ultra-low-radioactive materials and keep them clean for detector targets and for detector construction. The neutron background is one of the most critical backgrounds because the interaction of neutrons with a target is indistinguishable from the scattering of DM particles. So, to protect the sensitive target of the detector from cosmogenic neutrons [2] and the neutrons produced in (α, n) reactions [3], it is necessary to take additional measures.

It is assumed that the Dark Side-20K experiment, also focused on the direct DM search, will be able to accumulate a huge exposure and achieve a sensitivity of 7.4 × 10^−48^ cm^2^. This result can be a big fundamental breakthrough on the way to the development of direct DM search technology using liquid argon (LAr) as the recoil target [4].

Detectors with ultra-low background radioactivity, low energy thresholds, and high fiducially volumes will enable the next generation of discoveries in astroparticle physics. To keep these detectors ultra-low background, it is mandatory for most of them to be installed in underground low background laboratories to keep them from cosmic muons and cosmogenic activation.

An increase in the sensitive fiducial volume of the detector leads to an increase in its mass and size, which becomes a challenge in terms of the availability of the required amount of ultralow background materials, structural materials first of all, and the requirements for the size of experimental halls in underground laboratories. There are few structural industrial materials historically used, such as copper, stainless, and titanium alloys. The advantages of ultralow background titanium were shown and a proposal to use it as structural material for the cryostat in the detectors for DM search was made for the LUX and DS-50 experiments [5,6]. The idea was successfully confirmed by LZ Collaboration [7].

However, it is important to notice that the structural GRADE-1 titanium used for the LZ cryostat was obtained via pre-selection strategy. This strategy is usually used now and consists in sampling low background material from a large lot, which means that there is no guaranteed repeatability. Thus, the need for large batches of ultra-low background materials, on the scale of tens of tons, makes it important to develop industrial technologies for the production of ultra-low background materials with a level of residual contamination of ^238^U and ^232^Th less than 1 mBq/kg, which means U/Th concentrations below 0.1 ppb and structural titanium is the material which is very promising from this point of view.

Following the previously conducted investigations [8], we carefully studied the theory and practice of the Kroll process to understand its applicability to the production of ultra-pure Ti. It was shown that the use of titanium tetrachloride (TiCl_4_) makes it possible to reduce the amount of radioactive impurities. The inductively coupled plasma mass-spectrometry (ICP-MS) analysis of TiCl_4_ proves that the purity of the starting material purchased from the Component Reactiv LTD (Russia) is below 0.02 ppb for U and Th (Table 1). Thus, the Mg involved in the Kroll process was investigated. It is shown that the use of low background Mg is required to obtain high-purity Ti. Highly purified Mg can be obtained by upgrading an already existing vacuum separation process or by using the Pidgeon process, which includes a step of vacuum sublimation.

## 2. The Role of Magnesium in the Production of High-Pure Titanium

One of the known and most widely used Ti-sponge production methods is the Kroll process, based on the thermal reduction of TiCl_4_ by Mg. A simplified process diagram is shown in Figure 1. A metallothermic process at ~1000 °C with Mg is used to produce Ti sponges using purified TiCl_4_, which is pure in terms of radioactive impurities. Argon is pumped into the metallothermic reactor to prevent the Ti-sponge from oxygen and nitrogen contamination. Mg reacts with TiCl_4_ to produce liquid MgCl_2_ and ahard, porous Ti sponge flakes fall down to the bottom of the reactor and form, from bottom to top, the Ti-sponge block. The Ti sponge remains solid because the melting point of Ti is higher than that of the reaction. A vacuum separation process follows the metallothermic process to remove residual Mg and MgCl_2_ from the fresh Ti-sponge block.

The analysis of ICP-MS of various fractions of MgCl_2_ and Mg distributed over the retort showed that the final stage of vacuum separation can serve as a source for high-purity Mg. Thus, Mg can be used in the following metallothermic process to produce low background Ti sponges.

Vacuum separation is based on different vapor pressures of Ti, Mg, and MgCl_2_. Thus, the boiling points at atmospheric pressure for Ti, Mg, and MgCl_2_ are 3260, 1107, and 1417 °C, respectively. However, the maximum temperature on the retort wall should not exceed 1085 °C. At this temperature, iron noticeably interacts with Ti to form a fusible compound—eutectic. Therefore, for more complete removal of Mg and MgCl_2_ and lowering the temperature of the process, sublimation is carried out under high vacuum. In addition, the favorable conditions of vacuum distillation, which allow operation at lower temperatures than under normal conditions, facilitate the choice of material for the heating device [9].

## 3. Materials and Methods

### Impurity Determination by ICP-MS

To analyze chemical purity of Mg preparations, we used inductively coupled plasma mass-spectrometry (ICP-MS). Samples was transferred of a solid sample to liquid phase. Mg samples (~1 g) were dissolved in hydrochloric acid (7N) purified by surface distillation systems (BSB-929-IR, Berghof, Germany) in polytetrafluoroethylene (DAP-100, PTFE, Berghof, Germany) autoclaves using a SPEEDWAVEFOUR microwave decomposition system (Berghof, Germany). The dissolution product was transferred to a polypropylene (PP) test tube. The deionized water was produced by an Aqua-MAX-Ultra 370 Series setup (Young Lin Instruments, Republic of Korea) and had 18.2 MΩ cm electrical resistance and 99.999999 wt% purity (based on 68 elements). Then, the resulting solution was transferred to a PP test tube, and then the solution was brought by water. The solution thus prepared was analyzed by inductively coupled plasma mass-spectrometry (ICP-MS). Analytical measurements were carried out on a NexION300D inductively coupled plasma mass spectrometer (PerkinElmer Inc., Waltham, MA, USA).

The total quant method [11,12] was used for determination of 65 chemical elements’ concentrations. The quantitative analysis of Th and U was carried out using the “additives” method taking into account the concentration of the main (Mg matrix) elements in the analyzed solution. The standard solutions (PerkinElmer Inc.) were used for calibration. The optimized operating mode of the NexION300D spectrometer for impurity analysis of samples with Mg matrix element is presented in Table 2.

## 4. Results and Discussion

In previous studies [8], we showed that the Kroll process makes it possible to obtain low background Ti sponges if the starting materials are pure. The original industrial TiCl_4_ initially has a concentration of ^232^Th and ^238^U below the detection limit (Table 1). Therefore, it is necessary to pay attention to Mg.

In Russia, Mg is produced from enriched anhydrous carnallite by electrolysis (samples Mg-E#). Carnallite ores and a number of other deposits, along with the main mineral, contain sodium chloride, calcium and magnesium sulfates, and clay minerals. Obtaining artificial carnallite from these ores is carried out by the method of chemical enrichment. This method is based on the different joint solubility of MgCl_2_, KCl, and NaCl in water as a function of temperature. Artificial carnallite in its structure is a MgCl_2_KCl-6H_2_O. In an isolated system, MgCl_2_KCl-6H_2_O melts at a temperature of 168 °C in its own water of crystallization with decomposition into KCl crystals and an aqueous solution of MgCl_2_ [13].

Thus, the preparation of the initial carnallite raw materials for electrolysis should include the following operations:-Dehydration in the solid state with a gradual increase in temperature from 90 to 200–210 °C (dehydrated carnallite contains 2.5–5% water at the outlet of the unit);-Melting of dehydrated carnallite at 500 °C, followed by heating to 750–800 °C and holding at these temperatures in order to finally remove moisture and separate MgO (obtaining anhydrous carnallite).

Mg obtained by electrolysis was used to carry out the Kroll process. According to the results of ICP-MS, electrolytic Mg contains a high concentration of radioactive impurities, which does not allow its use (Figure 2). Mg needs to be further purified. One of the methods of purification is vacuum distillation. Laboratory studies have shown that the concentration of radioactive impurities decreases in the process of vacuum distillation (Figure 3).

In the process of vacuum distillation, not only the concentration of radioactive impurities decreases. As a result of Mg purification, it is possible to reduce the content of impurities such as Al, Ti, Cr, Mn, Fe, Cu, Pb, including radioactive impurities.

An alternative industrial method for obtaining Mg is the Pidgeon process [14], the final stage of which is vacuum sublimation. The Pidgeon process is based on the reduction reaction of fired dolomite with ferrosilicon, which can be summarized as follows:2MgO*^s^* + 2CaO*^s^* + (Si-Fe)*^s^* = 2Mg*^v^* + 2CaO × SiO_2_*^s^* + Fe*^s^*


According to this scheme, MgO is reduced to gaseous Mg, and CaO binds the resulting Si into a refractory CaO × SiO_2_. The final stage of the Pidgeon process is vacuum separation. In each retort, previously cleaned of residues (a mixture of dicalcium silicate with iron) of the previous cycle, briquettes of the obtained Mg are loaded. The retort is closed with a lid and pumped out to the exact pressure of 5–30 Pa. The average temperature of the process is 1150 °C. This temperature decreases somewhat when the retort is opened and reduced residues are removed from it or new batches of Mg are loaded and remain almost constant throughout the entire operation cycle. The average vacuum maintenance time in the retort is 9.5 h. The metallic Mg produced in China obtained in this way met the high requirements for radioactive impurity elements (Table 3). This shows the applicability of vacuum separation to the production of Mg with special requirements for radioactive purity. However, the concentration of the alpha decay isotope ^232^Th exceeds the required level of radioactivity of 0.02 ppb but is significantly better than Mg obtained by electrolysis.

An analysis of the metallothermic process allowed us to suggest that vacuum sublimation can be combined with the final stage of the Kroll process—vacuum separation. Vacuum separation is a vacuum sublimation of the MgCl_2_ and Mg contained in a Ti sponge. It can be assumed that with the modernization of the vacuum sublimation plant and the selection of technological processes, it will be possible to purify Mg with the vacuum separation process.

The analysis of Mg containing preparations at various stages of the technological process showed that the samples from China (Pidgeon process) and the MgCl_2_ obtained from this preparation after the Kroll process turned out to be the purest in terms of radioactive impurities (Figure 4).

Figure 4 presents the competing processes of the Th and U contamination of Mg in the process of Mg production: (1) an increasing of Th and U contamination in the electrolysis method and (2) a decreasing of Th and U concentrations in the metallothermic process. It can be observed how the contamination level rises at each subsequent stage of Mg production and reaches a maximum in commercial Mg ingots obtained by the electrolysis process.

For comparison, Figure 4 shows the results of the analysis of Mg obtained by the Pigeon process. One can see that the level of U contamination, which comes from the initial ore raw material, corresponds to the minimum levels. This shows the high efficiency of the Pigeon process in terms of the purity of the material obtained, except for the Th contamination.

Meanwhile, we observed the process of purification of Mg during its transition to MgCl_2_ in the metallothermic reaction shown by blue arrows. It was found that the recycled MgCl_2_ coming from the Ti sponge production process had U and Th concentrations which met the requirements of low radioactivity of 0.001 ppb (Appendix A). Thus, it would be eventually used for low-radioactive Mg reduction by the electrolysis process and further sublimation for production of low-radioactive Ti sponges (Figure 5).

## 5. Conclusions

The most common industrial method for obtaining magnesium is electrolysis. It was established that the concentration of alpha-decay U/Th isotopes in an electrolysis magnesium does not allow using it for the production of an ultra-low background Ti. The original magnesium must be purified before carrying out the Kroll process. It was established that vacuum sublimation is an effective method of magnesium deep purification from alpha-decay isotopes.

Thus, to obtain an ultra-low background Ti sponge, it is necessary and sufficient to exclude the use of commercial magnesium electrolysis. The optimal solution would be to add magnesium sublimation to the currently used technological process of Ti sponge production as shown in Figure 5. Intermediates should be constantly monitored for undesirable impurities by ICP-MS.

## Figures and Tables

**Figure 1 materials-15-08872-f001:**
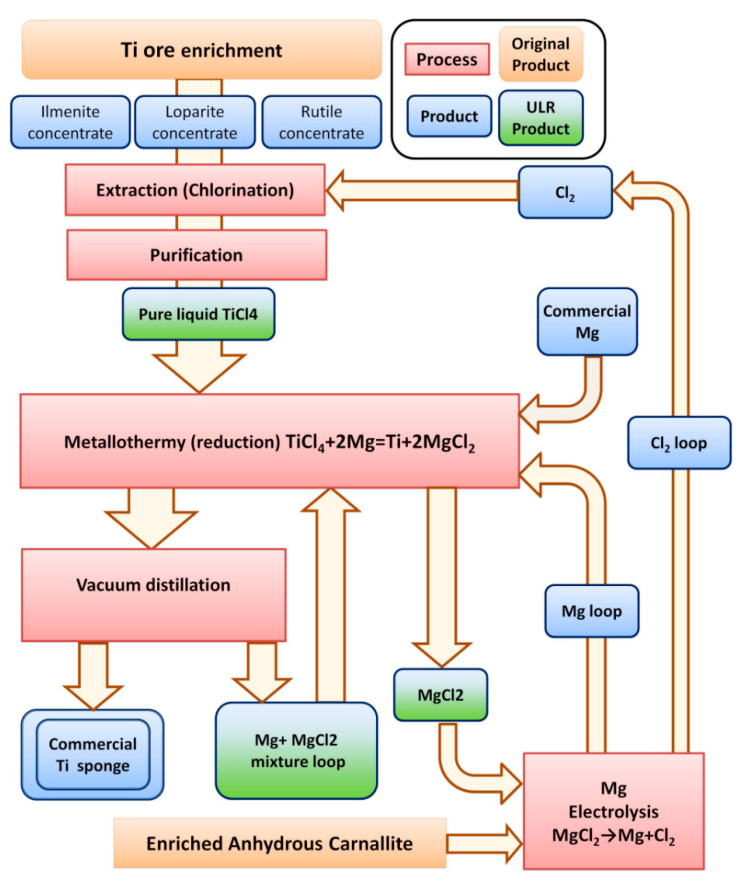
Simplified scheme of the industrial titanium sponge production process [8,10]. Ultra-low-radioactive (ULR) products discovered within the ULR titanium R&D are marked in green.

**Figure 2 materials-15-08872-f002:**
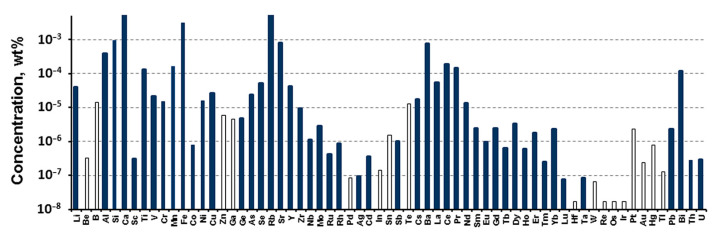
Impurity concentrations determined by ICP-MS in the Mg produced by electrolysis. Here and after the empty bars indicate the limits of determination (LD) of ICP-MS analysis.

**Figure 3 materials-15-08872-f003:**
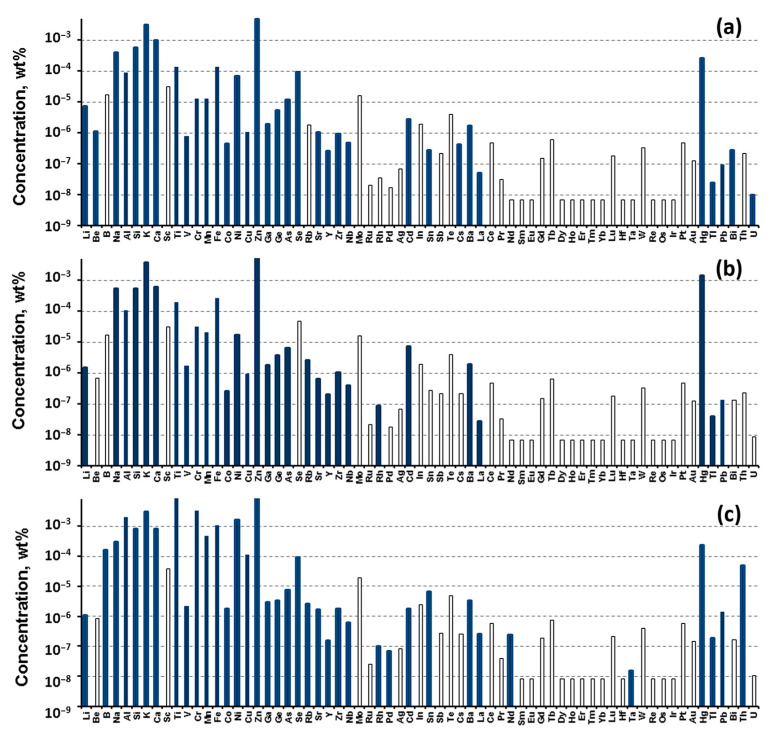
The ICP-MS results of Mg purified by vacuum distillation: (**a**) upper fraction called Mg-S-Top; (**b**) bottom fraction called Mg-S-Bot; (**c**) cube fraction called Mg-S-Cube (Appendix A).

**Figure 4 materials-15-08872-f004:**
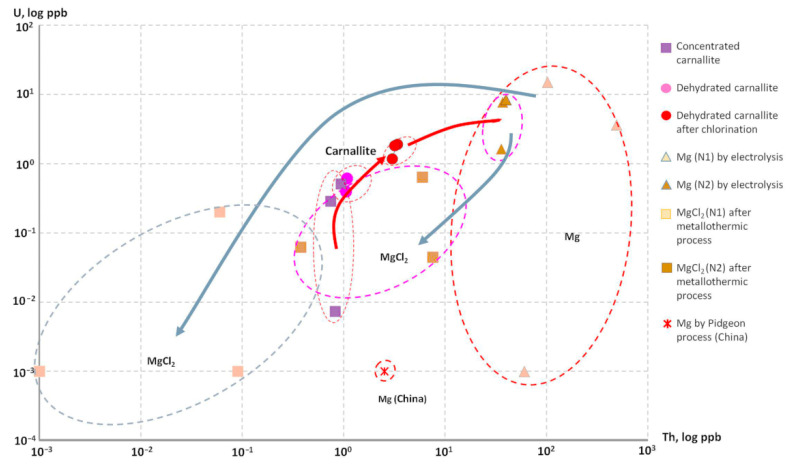
Distribution of U and Th in different samples of the materials used for Mg production via electrolysis process and Mg samples obtained by the various processes in comparison with MgCl_2_ outputs from Kroll process (Appendix A).

**Figure 5 materials-15-08872-f005:**
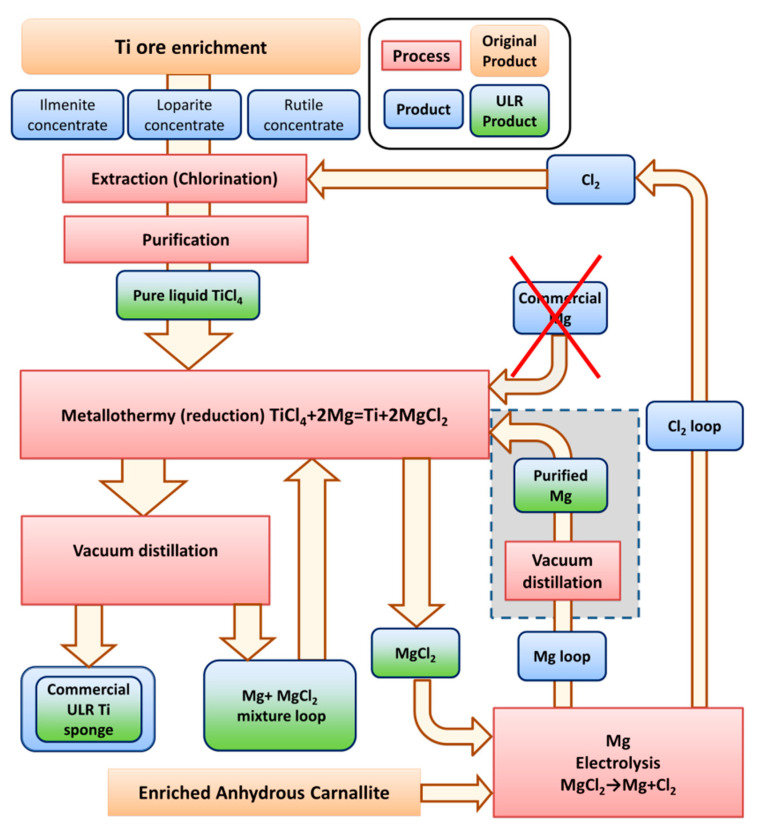
Modified scheme of ultra-low radioactivity Ti sponge production.

**Table 1 materials-15-08872-t001:** The ICP-MS results of titanium tetrachloride (Component Reactiv LTD, Russia).

	Th Concentration, ppb	U Concentration, ppb
TiCl_4_		
Sample 1	<0.02	<0.04
Sample 2	<0.02	<0.04
Sample 3	<0.02	<0.04

**Table 2 materials-15-08872-t002:** The operating mode of the NexION300D instrument for conducting impurity analysis of samples.

Nebulizer type	Concentric (Meinhard), PFA
Spray chamber	Scott double-pass chamber, PFA
Argon flow rate, L/min	
through the nebulizer	0.96
Plasma-forming	15
Auxiliary	1.2
Generator power, W	1450
Collision gas (He) flow rate, L/min	4.6
Number of scan cycles	8

**Table 3 materials-15-08872-t003:** The ICP-MS results of Mg.

Sample ID	Th Concentration, ppb		U Concentration, ppb	
	Content	Standard Deviation	Content	Standard Deviation
Mg obtained by electrolysis				
Mg-E1	37	2	7.7	0.4
Mg-E2	40	3	8.5	0.8
Mg-E3	36	3	1.6	0.2
Mg-E4	52	4	7.8	0.4
Mg after sublimation				
Mg-S-Top	10	3	0.04	0.01
Mg-S-Bot	2.9	0.9	0.9	0.3
Mg-S-Cube	1379	108	0.22	0.02
Mg obtained by Pidgeon process				
Mg-China	2.52	0.73	0.001	0.0005

## Data Availability

Not applicable.

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
