# Peer review of "Role of Magnesium in Ultra-Low-Radioactive Titanium Production for Future Direct Dark Matter Search Detectors"

_materials, 2022, doi:10.3390/ma15248872_

Round 1

Author Response

Dear Reviewers

Thank you for the fruitful comment. We took into consideration all the remarks.

We believe that it will enhance the article .

Reviewer 2 Report

1.Line 18. Please, use the term Dark Matter, not the Dark Metter.

2. Line 21, " It was found out that when transformed to MgCl2 in the Kroll process Mg was purified from U and Th. The further MgCl2 reduction and sublimation makes it possible to produce low-radioactive Ti sponge. " I suppose that there should be a comma instead of a point. It seems to be more reasonable to wright: « It was found out that when transformed to MgCl2 20 in the Kroll process Mg was purified from U and Th, the further MgCl2 reduction and sublimation makes it possible to produce low-radioactive Ti sponge.»

3. Line 48. It will be good to explain the “LAr” abbreviation in the manuscript.

4. Line 50? It will be interesting to know weather the problem may be solved with extremely pure Ti only. What about other materials necessary for the device, what are the conditions for the place on the Earth where it is possible to build it ? Some comments are strongly desirable .

5. Line 123. May be Ti sponge instead of Ti spongy?

Author Response

Dear Reviewers

Thank you for the fruitful comment. We took into consideration all the remarks.

We believe that it will enhance the article ..
